# CausalAF: Causal Autoregressive Flow for Safety-Critical Scenes Generation

## Abstract

Goal-directed generation, aiming for solving downstream tasks by generating diverse data, has a potentially wide range of applications in the real world. Previous works tend to formulate goal-directed generation as a purely data-driven problem, which directly approximates the distribution of samples satisfying the goal. However, the generation ability of preexisting work is heavily restricted by inefficient sampling, especially for sparse goals that rarely show up in off-the-shelf datasets. For instance, generating safety-critical traffic scenes with the goal of increasing the risk of collision is critical to evaluate autonomous vehicles, but the rareness of such scenes is the biggest resistance. In this paper, we integrate causality as a prior into the safety-critical scene generation process and propose a flow-based generative framework – *Causal Autoregressive Flow (CausalAF)*. CausalAF encourages the generative model to uncover and follow the causal relationship among generated objects via novel causal masking operations instead of searching the sample only from observational data. Extensive experiments on three heterogeneous traffic scenes illustrate that *CausalAF* requires much fewer optimization resources to effectively generate goal-directed scenes for safety evaluation tasks.

## 1 Introduction

Deep generative models (DGMs) have shown their powers for data generation in several domains. Recently, people have been weary of random generation and turned to generating goal-directed samples useful for downstream tasks. Standing on the top of successful DGMs, goal-directed generation demonstrates potentiality in molecule [32] and natural language [26] areas, which is usually formulated as shifting the generative distribution to satisfy specific goals.

One typical application of goal-directed generation is generating traffic scenes, which is a universally acknowledged way to evaluate autonomous vehicles [31]. Rare but significant, safety-critical scenes are extraordinarily important for the evaluation. Taking the *safety-critical* scene as a goal, such a generation task is challenging since we need to simultaneously consider scene realism to avoid conjectural scenes that will never happen in the real world, as well as the safety-critical level which are indeed rare compared with ordinary scenes. In addition, generating reasonable threats to vehicles' safety can be inefficient if the model purely relies on the correlation of observation, as the safety-critical scenes are rare and follow certain fundamental physical principles.

Existing work [12] searches in the latent space of generative model to build scenes that satisfy downstream requirements. The biggest challenge is that ordinary scenes may dominate the latent space while safety-critical samples are ignored as "outliers". Another approach [36] is to retrain the model during the searching to avoid forgetting the high-quality but rare data. However, the efficiency could still be unacceptably low due to the sparsity of qualified samples. In contrast, humans are good at abstracting the causation beneath the observations with prior knowledge, which lights up a new direction towards causal generative models.

In this paper, we build a goal-directed generative model with causal priors that are accessible in many applications. We model the causality as a directed acyclic graph (DAG) named causal graph (CG) [29]. To facilitate CG in the downstream tasks, we propose the Behavioral Graph (BG), which can be regarded as instances of CG [16], for interactive and dynamic scenes representation. The graphical representation of both graphs makes it possible to use the BG to unearth the causality given by CG. We propose the first generative model that integrates causation into the graph generation task

and name it *CausalAF*. To connect BG and CG at the graph level, we propose two types of causal masks – Causal Order Masks (COM) and Causal Visibility masks (CVM). COM modifies the node order for node generation, and CVM removes irrelevant information for edge generation.

For a better explanation, we consider a running example of a traffic scene. When the vision of the autonomous vehicle $a$ is clear, $a$ can easily see the pedestrian $c$ crossing the road then decelerate in advance. However, if another vehicle $b$ is parked in the middle between $a$ and $c$, the vision of $a$ will be blocked, making $a$ have less time to brake and more likely to collide $c$. This example may take autonomous driving vehicles millions of hours to collect [13], which is challenging for real-world applications. However, when we use a generative model to create such a scene, it will not consider the causality but try only to memorize the location of all objects then generate adversarial examples [15]. Consequently, the generated scene may not cause any risk if the objects are slightly different.

Overall, we show the diagram of goal-directed generation with *CausalAF* in Fig. 1 and we summarize our contributions below:

- We proposed a causal generative model named *CausalAF* that integrates causal graphs and temporal graphs for safety-critical scene generation.

- We designed two novel mask operators to reliably integrate causation order and causation visibility into the flow-based generation procedure.

- We showed *CausalAF* demonstrates dramatic improvement in efficiency and generalizability on three standard traffic settings compared with purely data-driven goal-directed baseline.

## 2 REPRESENTATION OF CAUSATION AND SCENES

Our *CausalAF* is built upon the relation between the CG and the BG. We start by introducing the definition of these two types of graphs and the autoregressive generation process of the BG.

### 2.1 CAUSAL GRAPH AND BEHAVIORAL GRAPH

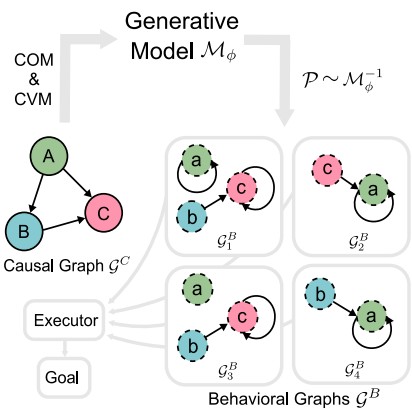

Figure 1: Diagram of proposed *CausalAF* framework.

The causal graph is defined over $m$ random variables $\{x_1, ..., x_m\}$. The variables in this vector forms a DAG $\mathcal{G}^C = (V^C, E^C)$. $V^C \in \{0, 1\}^{m \times n}$ is the node matrix and $E^C \in \{0, 1\}^{m \times m}$ is the adjacency matrix with $m$ nodes in $n$ types. Each node $i$ is associated with a random variable $x_i$. Each edge $(i, j)$ represents a causal relation from variable $x_i$ to $x_j$. For a DAG, there exists a (not necessarily unique) causal order of the nodes, such that the cause variable precedes the effect variable: $p(x_1, ..., x_n) = \prod_{j=1}^{n} p_j(x_j \mid \mathbf{pa}(x_j))$, where $\mathbf{pa}(x_j)$ represents the parent nodes for variable $x_j$. In this work, we assume $\mathcal{G}^C$ is fully accessible with human knowledge and experience for certain tasks.

We then define the Behavioral Graph $\mathcal{G}^B$ to represent objects in a dynamic and interactive scene. According to **Definition** 1, $\mathcal{G}^B$ works as a high-level planner for objects and controls their behaviors in the physical scene with interpretable edge meanings. A self-loop edge $(i, i)$ represents that one object takes one action irrelevant to other objects (e.g., a car goes straight or turns left with no impact on other road users), while other edges $(i, j)$ means object $i$ takes one action related to object $j$ (e.g., a car $i$ moves towards a pedestrian $j$). The edge attributes represent the properties of actions. For instance, the attribute $[x, y, v_x, v_y]$ of one edge represents the 2-d position and velocity for agent nodes.

**Definition 1** *(**Behavioral Graph**) Suppose there are $n$ types of nodes and a scene have $m$ objects. Then the Behavioral Graph $\mathcal{G}^B = (V^B, E^B)$ contains a node matrix $V^B \in \mathbb{R}^{m \times n}$ representing the categories of objects and an edge matrix $E^B \in \mathbb{R}^{m \times m \times (h_1 + h_2)}$ representing the sequential interaction between objects, where $h_1$ is the number of edge types and $h_2$ is the dimension of edge attributes.*

## 2.2 BEHAVIORAL GRAPH GENERATION WITH AUTOREGRESSIVE FLOW

Considering the directed acyclic nature of $\mathcal{G}^C$, we incorporate autoregressive flow models (AF) [18], which is a type of DGMs that sequentially generate nodes based on their predecessors to generate $\mathcal{G}^B$. It uses an invertible and differentiable transformation $f$ to convert the observations $\boldsymbol{x}$ to a latent variable $\boldsymbol{z}$ that follows a base distribution $p_0(\boldsymbol{z})$ (e.g., Normal distribution). According to the change of variables theorem, we can obtain $p_{\boldsymbol{x}}(\boldsymbol{x}) = p_0(f^{-1}(\boldsymbol{x})) \left| \det \frac{\partial f^{-1}(x)}{\partial \boldsymbol{x}} \right|$. To increase the representing capability, we repeatedly substitute the variable for the new variable $z_i$ and eventually obtain a probability distribution of $\boldsymbol{x}$ whose log-likelihood can be written as:

$$\log p(\boldsymbol{x}) = p_0(\boldsymbol{z}_0) - \sum_{i=1}^{K} \log \left| \det \frac{df_i}{d\boldsymbol{z}_{i-1}} \right| \tag{1}$$

In AF models, the transformation $f$ construct $\boldsymbol{x}$ in a sequential way, which is naturally consistent with the construction of $\mathcal{G}^C$. To implement the function invertible $f$, we build a model $\mathcal{M}_\phi$ parametrized by $\phi$. The inverse of $\mathcal{M}_\phi$, denoted as $\mathcal{M}_\phi^{-1}$, can be used to sample new data from Gaussian noises: $\boldsymbol{x} = \boldsymbol{z}_K = f_K^{-1} \circ f_{K-1}^{-1} \circ \cdots \circ f_0^{-1} = \mathcal{M}_\phi^{-1}(\boldsymbol{z}_0)$, where $\circ$ means the composition of two functions and $\boldsymbol{z}_0 \sim \mathcal{N}(\boldsymbol{0}, \boldsymbol{I})$. Let $V_{[i,:]}^B$ and $E_{[i,j,:]}^B$ represent the node $x_i$ and edge $(i, j)$ of $\mathcal{G}^B$ sampled from Gaussian distribution

$$V_{[i,:]}^B \sim \mathcal{N}\left(\mu_i^v, (\sigma_i^v)^2\right) = \mu_i^v + \sigma_i^v \odot \epsilon, \quad E_{[i,j,:]}^B \sim \mathcal{N}\left(\mu_{i,j}^e, (\sigma_{i,j}^e)^2\right) = \mu_{i,j}^e + \sigma_{i,j}^e \odot \epsilon \tag{2}$$

where $\odot$ denotes the element-wise product. $\epsilon$ follows a Normal distribution $\mathcal{N}(\boldsymbol{0}, \boldsymbol{I})$ and $[:]$ represents all elements in one dimension. In (2), variables $\mu_i^v, \sigma_i^v, \mu_{i,j}^e$, and $\sigma_{i,j}^e$ are obtained from $\mathcal{M}_\phi$:

$$\mu_i^v, \sigma_i^v = \mathcal{M}_\phi(V_{[0:i-1]}^B, E_{[0:i-1,:]}^B), \quad \mu_{i,j}^e, \sigma_{i,j}^e = \mathcal{M}_\phi(V_{[0:i]}^B, E_{[0:i,0:j-1]}^B) \tag{3}$$

where $[0 : i]$ represents the elements from index 0 to index $i$. According to (3), the generation of the current node depends on all previous nodes and edges. Then the edges between current node and previous nodes are generated. Eventually, $E^B$ will be an upper-triangular matrix since only the latter generated nodes have edges pointed to formerly generated nodes. This process is illustrated in Fig. 2.

## 3 CAUSAL AUTOREGRESSIVE FLOW (CAUSALAF)

Transferring the prior knowledge from $\mathcal{G}^C$ to $\mathcal{G}^B$ can be implemented by increasing the similarity between them. However, this similarity is not easy to calculate because it includes the directions between nodes and the input information of nodes. To solve this problem, we propose the *CausalAF* model with two causal masks, i.e., Causal Order Masks (COM) and Causal Visible Masks (CVM), that make the generated $\mathcal{G}^B$ follow the causal information given in $\mathcal{G}^C$. Particularly, COM is designed for regulating the order of the node generation, and CVM dynamically masks out irrelevant information during the edge generation.

**Causal Order Masks** The order is vital during the generation of $\mathcal{G}^C$ since we must ensure the cause is generated before the effect. To achieve this, we maintain a priority queue $\mathbb{Q}$ to store the valid node types for the current step. $\mathbb{Q}$ is initialized with $\mathbb{Q} = \{x_i | \mathbf{pa}(x_i) = \emptyset\}$, which means all nodes that do not have parent nodes are valid at the beginning. Then, in each node generation step, we update $S$ by removing the generated node $x_i$ and adding the child nodes of $x_i$. Notice that one node could have multiple parents; thus, we consider one node valid only if all of its parents have been generated. To encourage the model to generate nodes that satisfy the causal order, we use $\mathbb{Q}$ to create a $k$-hot mask $M^o(\mathcal{G}^C) \in \mathbb{R}^n$, where the element is set to 1 if it is corresponding to a valid node. Then, the type of next node $x_i$ will be obtain by $v_i = \arg\max(M^o(\mathcal{G}^C) \odot \text{softmax}(V^B[i, :]))$, where $V^B[i, :]$ is the original node matrix obtained from $\mathcal{M}_\phi$ for node $x_i$. Intuitively, this mask reduces the probability of the invalid node types to 0 to ensure the generated node follows the correct order.

**Causal Visible Masks** Ensuring a correct causal order is still insufficient to represent the causality, which will be discussed in the later experiments. Thus, we further propose another type of mask called CVM. COM serves as a precondition for CVM in that it guarantees the existence of one node's parents before this node is ready to be generated. Otherwise, one node may lose prior information without knowing its causes.

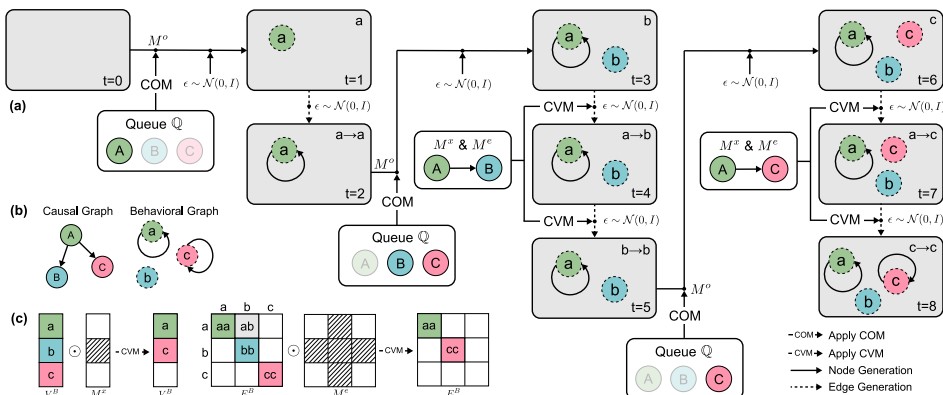

Figure 2: **(a)** The generation process of a Behavioral Graph. **(b)** The causal graph and Behavioral Graph used in the example of **(a)**. **(c)** The explanation of CVM when generating edges for $c$, where irrelevant node $b$ is masked out in both $V^B$ and $E^B$.

At the step of generating edges for node $x_i$, we maintain the current generated graph with $\mathcal{G}^B(t) = (V^B(t), E^B(t))$, where $t$ is the index for current step. Then, CVM is implemented with $M^x(\mathcal{G}^C) \in \mathbb{R}^{m \times n}$ and $M^e(\mathcal{G}^C) \in \mathbb{R}^{m \times m \times (h_1+h_2)}$ that satisfy

$$M^x(\mathcal{G}^C)[j, :] = 0, \;\; M^e(\mathcal{G}^C)[:, j, :] = \mathbf{0}, \;\; M^e(\mathcal{G}^C)[j, :, :] = \mathbf{0}, \;\; \forall \{j \mid x_j \notin \mathbf{pa}(x_i)\} \quad (4)$$

With these two masks, we can update $\mathcal{G}^B(t)$ before using it for next step by

$$V^B(t) \leftarrow V^B(t) \odot M^x(\mathcal{G}^C), \;\; E^B(t) \leftarrow E^B(t) \odot M^e(\mathcal{G}^C) \quad (5)$$

We illustrate an example of CVM in (c) of Fig. 2. Assume we are generating edges for node $c$. We need to remove node $b$ since node $B$ does not have edges to node $C$. After applying $M^x(\mathcal{G}^C)$ and $M^e(\mathcal{G}^C)$, we move the features of node $c$ to the previous position of $b$. This permuting operation is important since the autoregressive model is not permutation invariant.

**Goal-directed Optimization**  We then discuss the training of *CausalAF*. The target of goal-directed generation is to create samples satisfying a given goal, which is formulated as an optimization over objective function $\min_\phi \mathbb{E}_{\mathcal{G}^B \sim M_\phi^{-1}}[\mathcal{L}_g(\mathcal{G}^B)]$. Usually, the objective $\mathcal{L}_g$ contains non-differentiable operators (e.g., complicated simulation and rendering), thus we have to utilize black-box optimization methods to solve the problem. We consider a policy gradient algorithm named REINFORCE [39], which estimates the gradient from samples by

$$\nabla_\phi \mathcal{L}_g(\mathcal{G}^B) = \mathbb{E}_{\mathcal{G}^B \sim M_\phi^{-1}}[\nabla_\phi \log M_\phi(\mathcal{G}^B)\mathcal{L}_g(\mathcal{G}^B)] = \frac{1}{N}\sum_{i=1}^{N}(\nabla_\phi \log M_\phi(\mathcal{G}_i^B)\mathcal{L}_g(\mathcal{G}_i^B)) \quad (6)$$

where $N$ is the number of samples used for each iteration. Overall, the entire training algorithm is summarized in **Algorithm** 1 in Appendix.

## 4 EXPERIMENT

We evaluate *CausalAF* using three top pre-crash traffic scenes defined in [27] and [38]. The benefit of the experimental setting is that humans usually have good intuitions of traffic scenes to examine the results. However, our empirical results show that it may not be trivial for the generative models to learn the underlying causality given the observational data, even if such causality seems understandable to humans. Particularly, we conduct a series of experiments to answer the question: whether there is a significant benefit to integrate causation into the generative models? We found that *CausalAF* outperforms the baseline and the advantages can be mainly attributed to the causation introduced by COM and CVM that eliminates irrelevant variables.

**Simulator for typical Scenes**  We consider three safety-critical traffic scenes (shown in Fig. 3) that have clear causality. The $\mathcal{G}^C$ for each scenario is displayed on the upper right of the scene. These $\mathcal{G}^C$ are not necessarily unique for the scene, while they just hypothesize the potential causation.

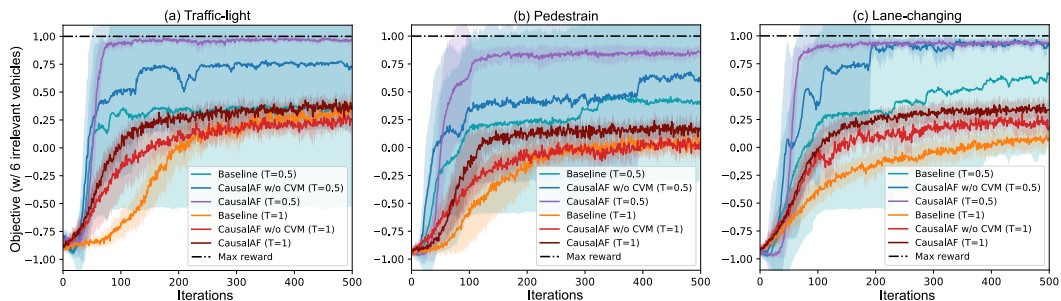

Figure 3: Three causal traffic scenes used in our experiments with corresponding causal graphs

Figure 4: The training objective $\mathcal{L}_g(\mathcal{G}^B)$ of three scenes under two temperature settings.

The details of these scenes can be found in the Appendix B. We implement these scenes in a 2D simulator, where all agents have radar sensors and dynamics. During the experiments, the goal-directed generative model firstly samples an $\mathcal{G}^B$. Then, the physical properties (e.g., position and velocity) defined in the generated $\mathcal{G}^B$ is executed in the simulator to create sequential scenes. After the execution, the simulator outputs the objective function $L_g(\mathcal{G}^B)$ as the simulation result.

Our goal is to generate risky scenarios that make collision happen for node $A$. Therefore, we set the object function to be a very sparse function: $\mathcal{L}_g(\mathcal{G}^B) = 1$ only if $\mathcal{G}^B$ causes collisions. Since generating goal-directed scenes is a new task, there are no existing methods to compare. We implement a baseline model with exactly the same structure as CausalAF without considering the causation during generation to represent data-driven generative models. We also compare with a model without CVM to conduct ablation studies.

**Results and discussion** We show the training objectives of three scenes in Fig 4. Notice that there are two temperatures $T = 0.5$ and $T = 1.0$ for all methods, which is use to control the sampling variance $\epsilon \sim \mathcal{N}(0, T)$. A large temperature provides strong exploration but also causes slow convergence. In all three scenes, *CausalAF* outperforms baseline, and the gap is more significant under $T = 1.0$ setting than $T = 0.5$. The reason could be that the new node heavily depends on previously generated nodes in the autoregressive generation of $\mathcal{G}^B$. The baseline has more noisy and irrelevant relations between nodes; therefore, it is less efficient to find the scenes that achieve $\mathcal{L}_g$. In addition, a strong exploration makes the irrelevant information have more influence on the baseline. In contrast, our *CausalAF* ignores the insignificant information and focuses on the causation that helps with the goal. We also find that *CausalAF* without CVM performs a little worse than *CausalAF*, which validates our hypothesis that COM may not be powerful enough to represent causality.

## 5 CONCLUSION

This paper proposes a causal generative model that generates safety-critical scenes with causal graphs obtained from humans prior. To incorporate the graphical structure of causal graphs, we design a novel scene representation called the Behavioral Graph. The autoregressive generation process of BG makes it possible to inject the causation via regulating the generating order and modifying the graph connection. By introducing causation into generative models, we are able to efficiently create rare scenes that might be difficult to find, such as safety-critical traffic scenes. Our method outperforms the baseline in terms of efficiency and performance on three scenes that have clear causation. One limitation of this work is that the causal graph, usually summarized by humans, is assumed to be always correct. Automatically discovering the causal graph will be the future direction.

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

## A  ALGORITHM SUMMARY

The training algorithm of our CausalAF method is summarized below:

---
**Algorithm 1:** Training process of CausalAF

---
**Input:** Dataset $\mathcal{D}$, Causal Graph $\mathcal{G}^C$, Goal $\mathcal{L}_g$, Learning rate $\alpha$, Maximum node number $m$
**Output:** The trained model $\mathcal{M}_\phi$

1   Initialize $\mathcal{M}_\phi$ by maximizing (1) on $\mathcal{D}$
2   **while** *not converged* **do**
      // Sample an BG from model $\mathcal{G}^B \sim M_\phi^{-1}$
3     **for** $i < m$ **do**
4       Sample a node $V^B[i,:]$ by (2)
5       Calculate $M^o(\mathcal{G}^C)$ for COM and apply (3) to get the node type $v_i$
6       Calculate $M^x(\mathcal{G}^C)$ and $M^e(\mathcal{G}^C)$ for CVM by (4)
7       **for** $j < i$ **do**
8         Apply CVM to node matrix $V^B$ and edge matrix $E^B$ by (5)
9         Sample an edge $E^B[i,j,:]$ by (2)
10      **end**
11    **end**
      // Learn model parameters
12    Calculate the likelihood $M_\phi(\mathcal{G}^B)$ of the sample
13    Execute $\mathcal{G}^B$ to get the goal objective $\mathcal{L}_g(\mathcal{G}^B)$
14    Update parameters with $\phi = \phi - \alpha\nabla_\phi\mathcal{L}_g(\mathcal{G}^B)$ by gradient estimated via (6)
15 **end**

---

## B  DETAILS OF EXPERIMENT SCENE

- **Traffic-light**. One potential safety-critical event could happen when the traffic light $T$ turns from green to yellow to give road right to an autonomous vehicle $A$. $R$ runs the red light, colliding with with $A$ perpendicularly. Here, $A$ node is the parent for both $T$ and $R$. $T$ is also a parent for $R$ because the risk vehicle follows the traffic light $T$.

- **Pedestrian**. A pedestrian $P$ and an autonomous vehicle $A$ are crossing the road in vertical directions. There also exists a static vehicle $S$ parked by the side of the road. Then a potentially risky scene could happen when $S$ blocks the vision of $A$ and $P$. In this scene, $A$ node is the parent for both $P$ and $S$. $S$ is also a parent for $P$ since $S$ determines the vision of $P$.

- **Lane-changing**. An autonomous vehicle $A$ takes a lane-changing behavior due to a static car $S$ parked in front of it. Meanwhile, a vehicle $R$ drives in the opposite lane. When $S$ blocks the vision of $A$, then $A$ is likely to collide with $R$. In this scene, we make $A$ node as the parent for both $R$ and $S$. $S$ is also a parent for $R$ since the $S$ determines the vision of $P$.

## C  RELATED WORK

### C.1  GOAL-DIRECTED GENERATIVE MODELS

DGMs, such as Generative Adversarial Networks [14] and Variational Auto-encoder [22], have shown powerful capability in randomly data generation tasks [5]. Thanks to the boom of diverse DGMs, goal-directed generation methods are widely used in many applications [26]. One line of research leverages conditional GAN [24] and conditional VAE [33], which take as input the conditions or labels during the training stage. Another line of research injects the goal into the model after the training. [12] proposes a latent space optimization framework that finds the samples by searching in the latent space. This spirit is also adopted in other fields: [25] finds the molecules that satisfy specific chemical properties, [1] searches in the latent space of StyleGAN [20] to obtain targeted images.

Recent works combine the advantages of the above two lines by retraining the generative model during the search. To expand the area of the desired region in the latent space, [36] iteratively updates the high-quality samples and retrains the model weights. [32] pre-trains the generative model and optimize the sample distribution with reinforcement learning algorithms. This paper enhances the generalizability and efficiency by leveraging causation graphs so that it is applicable to rare safety-critical scenes.

## C.2 SAFETY-CRITICAL TRAFFIC SCENE GENERATION

Traditional traffic scene generation algorithms sample from pre-defined rules and grammars, such as probabilistic scene graphs [30] and heuristic rules [11]. In contrast, DGMs [6, 35, 7, 8] are recently used to learn the distribution of objects to construct diverse scenes. There are two lines of work. One is to directly search for the adversarial scenes. [42] modifies the light condition. [3, 40, 19] manipulate the pose of objects in traffic scenes. [37, 2] adds objects on the top of existing vehicles to make them disappear, [34] creates a ghost vehicle by adding an ignorable number of points, and [10] generates the layout of the traffic scene with a tree structure integrated with human knowledge. Another line of research generates the risky scenes while also considering the likelihood of occurring of the scenes in the real world, which requires a probabilistic model of the environment. [43, 28, 4] used various importance sampling approaches to generate risky but probable scenes. [8] merges the naturalistic and collision datasets with conditional VAE to generate near-misses. [9] uses reinforcement learning to search for risky cyclist encounters for victim cars with a penalty of rarity. Compared with purely probabilistic methods, *CausalAF* method may have better generalization, data efficiency, and statistically robust against sparse data as it not only learns Bayesian models but also capture the causation of collisions.

## C.3 CAUSAL GENERATIVE MODELS AND REPRESENTATION LEARNING

The research of causality, mainly described with probabilistic graphical models-based language [29], is usually divided into two aspects: causal discovery tries to find the underlying mechanism from the observational and interventional data. In contrast, causal inference extrapolates the given causality to solve new problems. Discovering the causal graph has been prevalent for several decades. [45] proposed a flexible and efficient RL-based method to search over the DAGs space for the best causal graph that fits the dataset. A toolbox named NOTEARs is proposed in [44] to learn causal structure in a fully differentiable way, which drastically reduces the complexity caused by combinatorial optimization. [17] show the identifiability of learned causal structure from interventional data, which is obtained by manipulating the causal system under interventions.

Recently, causality has been introduced into DGMs to learn the cause and effect with representation learning. CausalGAN [23] captures the causation between labels by training the generator with the causal graph as a prior, which is very similar to our setting. In CausalVAE [41], the authors disentangle latent factors by learning a causal graph from data and corresponding labels. Previous work CAREFL [21] also explored the combination of causation and autoregressive flow-based model and is used for causal discovery and prediction tasks.

