# OpenReview forum: "CausalAF: Causal Autoregressive Flow for Safety-Critical Scenes Generation"
_ICLR.cc/2022/Workshop/OSC — Submitted to ICLR2022 OSC _

### Official Review · Reviewer_qNn2 · 2022-03-11
**Not a very accessible paper but interesting idea**

**Rating:** 2
**Confidence:** 1

**Review:**

If I understand correctly this paper proposes a model which allows goal directed samples (goal defined by a specific behavioural graph) while constraining the model to respect a given causal graph. The method uses flows, together with masks over order or visibiliy to model the edges of the graphs and obtain a distribution which is both easy to sample from as well as respecting the causal constraints and goal.
The method is demonstrated to work well producing samples of driving scenes which are consistent with the specified behavioural graph.

It was a bit hard to follow the paper and though it's nicely presented it's not accessible to someone who doesn't know much about causality.

I must say this is a bit out of my area of expertise but the paper seems interesting and I imagine people more well versed in causality would be interested in discussing this at the workshop.

---

### Official Review · Reviewer_SvXc · 2022-03-14
**CausalAF: Causal Autoregressive Flow for Safety-Critical Scenes Generation**

**Rating:** 1
**Confidence:** 1

**Review:**

Summary:
This work tackles the challenge of goal-directed scene generation in the context of generating safety-critical traffic scenes with a high risk of collision. The authors use the causal graphs of three high-risk traffic scenes to condition the scene generation process. This is done by generating a “behavioural graph” BG which represents a single scene, conditioned on an underlying causal graph CG. The generation of a BG follows the causal structure of an underlying CG via the use of masking operations. This generation process uses bijective transformations. This is used together with REINFORCE to train the model using a simulator which provides a positive reward only if the generated BG leads to a collision.

Clarity:
I found the paper a bit difficult to follow, sections 2 & 3 could be made more accessible to newcomers, and there are some broad / vague statements that would benefit from further elaboration. For example:
- “Recently, people have been weary of random generation and turned to generating goal-directed samples useful for downstream tasks.” → This is a bit of an unusual statement, at least without further clarification, given that there is still significant research into unconditional image generation.
- “We propose the first generative model that integrates causation into the graph generation task […]” → This is quite a broad claim and I am not entirely convinced that it is accurate in this form (e.g., [1]).
- It is never clearly stated *why* a flow-based formulation is used. In my understanding from Algorithm 1 in the appendix, this is to make log p(x) tractable so that REINFORCE can be applied. This should be explained in the main body of the paper.
- The notation in section 2.2 appears to be inconsistent, with z=f(x) in some cases and x=f(z) in others.
- It would help the reader to explain the intuition behind the CVMs before providing the mathematical description.
- It is not clear from the manuscript whether separate models are trained for each CG or whether a single models is trained on all three CGs.
- Algorithm 1, line 1 states "Initialize M_phi by maximising (1) on D" → There is no definition of what constitutes the dataset D and this step is not explained anywhere in the text.

Technical correctness:
- In the causal graphs in Figure 3b and 3c, why is the autonomous vehicle A a parent node of the standing vehicle S? The autonomous vehicle should not exert any influence on the standing vehicle. The standing vehicle should limit the vision of the autonomous vehicle and impact its behaviour rather than the other way around?
- It is claimed that the proposed approach outperforms the baselines. This is supported via reward curves for the three scenarios, but no qualitative results are provided. It is therefore unclear, e.g., whether there is any diversity in the scenes that are generated by the models, or whether the models just generate one specific scene with some negligible amount of noise.
-  In my understanding, there is nothing that prevents the standard deviations of the BG vertices and edges from collapsing towards zero. Similarly, changing the temperature T merely scales the noise variable, which in turn can be arbitrarily re-scaled by the model?

Strengths:
- The approach of using causal graphs together with a flow-based model to generate “behavioural graphs” that represent a scene is very interesting.
- The paper is highly relevant to the workshop.
- The work is well-motivated from an application perspective.

Weaknesses:
- The paper is a bit hard to follow and there are some broad / vague statements that would benefit from further elaboration.
- There are some questions concerning technical correctness.
- No details are provided about the simulator that is used in this work.

Conclusion:
The direction of this work is both interesting and relevant to the workshop. However, there is quite a bit of scope for improving the presentation and I am concerned that the results are potentially misleading.

[1] “Unbiased Scene Graph Generation from Biased Training”, Tang et al., 2020

---

### Decision · Program_Chairs · 2022-03-21

**Decision:**

Reject

**Comment:**

As highlighted by the reviewers, this is an interesting contribution with high relevance to the workshop. The work is well-motivated and presents a novel contribution. As pointed out by both reviewers, however, the work would benefit from significant improvement in terms of clarity of the presentation to make it more accessible to a wider audience and there were some open questions around technical correctness. Given that there is likely insufficient time until the camera-ready deadline to address the reviewer feedback in full, we recommend submitting this work to a later venue.